# SFPFusion: An Improved Vision Transformer Combining Super Feature Attention and Wavelet-Guided Pooling for Infrared and Visible Images Fusion

**DOI:** 10.3390/s23187870

**Published:** 2023-09-13

**Authors:** Hui Li, Yongbiao Xiao, Chunyang Cheng, Xiaoning Song

**Affiliations:** International Joint Laboratory on Artificial Intelligence of Jiangsu Province, School of Artificial Intelligence and Computer Science, Jiangnan University, Wuxi 214122, China; yongbiao_xiao_jnu@163.com (Y.X.); chunyang_cheng@163.com (C.C.); x.song@jiangnan.edu.cn (X.S.)

**Keywords:** image fusion, Transformer, wavelet-guided pooling, global features, detail features

## Abstract

The infrared and visible image fusion task aims to generate a single image that preserves complementary features and reduces redundant information from different modalities. Although convolutional neural networks (CNNs) can effectively extract local features and obtain better fusion performance, the size of the receptive field limits its feature extraction ability. Thus, the Transformer architecture has gradually become mainstream to extract global features. However, current Transformer-based fusion methods ignore the enhancement of details, which is important to image fusion tasks and other downstream vision tasks. To this end, a new super feature attention mechanism and the wavelet-guided pooling operation are applied to the fusion network to form a novel fusion network, termed SFPFusion. Specifically, super feature attention is able to establish long-range dependencies of images and to fully extract global features. The extracted global features are processed by wavelet-guided pooling to fully extract multi-scale base information and to enhance the detail features. With the powerful representation ability, only simple fusion strategies are utilized to achieve better fusion performance. The superiority of our method compared with other state-of-the-art methods is demonstrated in qualitative and quantitative experiments on multiple image fusion benchmarks.

## 1. Introduction

Image fusion is the combination of multiple source images carrying complementary information to generate a high-quality fused image, which is used to make up for the limitation of incomplete image information obtained by a single type of sensor. As a task in the field of image processing, image fusion is applicable to various downstream tasks, including target detection [1,2], object tracking [3,4], and semantic segmentation [5].

Infrared image and visible image are two typical data sources studied in image fusion. Infrared image contains salient thermal infrared features, which are not affected by external conditions such as bad weather and poor illumination, but usually lack texture details [6]. Visible image contains more texture details and is suitable for human visual perception but very sensitive to illumination and weather conditions [7]. Therefore, infrared and visible image fusion methods are utilized to generate one image that contains both salient features and rich texture details.

In general, the key to infrared and visible image fusion is how to extract and fuse complementary features while reducing redundant information. In recent years, a large number of fusion algorithms are proposed, which can be roughly divided into non-deep learning and deep learning-based methods. Non-deep learning methods are mainly based on signal processing operations [8,9]. For example, multi-scale transform (MST) [10,11] can decompose the original image into components of different scales and extract multi-scale features. Then, an effective fusion strategy is designed to fuse the features. Finally, the image is reconstructed using the inverse transformation of feature extraction. In addition, the methods of non-deep learning also include sparse representation (SR) [12,13,14,15]-based methods, subspace [16,17]-based methods, and low-rank representation (LRR) [18,19,20,21]-based methods. Although the non-deep learning methods can synthesize satisfactory results, they still have some drawbacks: (1) manually designed fusion strategies cannot adapt to complex image fusion conditions and have poor generalization ability; (2) manual feature extraction has limitations in comprehensively capturing multi-modal images, which introduces noise and causes image distortion.

With the rise of deep learning, a lot of fusion methods based on deep learning are proposed to solve the above shortcomings [22]. Existing deep learning-based fusion methods can be divided into four categories: (1) convolutional neural network (CNN) [7,23]-based methods; (2) auto-encoder (AE) [24,25]-based methods; (3) generative adversarial network (GAN) [26,27]-based methods; and (4) Transformer [28,29]-based methods.

The CNN-based methods have excellent feature extraction capabilities, but the simple use of CNN cannot extract the deep and global features of source images. Therefore, in order to extract more deep features, the AE-based training strategy was introduced to image fusion task, which has strong generalization ability and can obtain richer image features. However, this kind of fusion method still needs complex fusion strategies to be manually designed. Moreover, given the powerful ability of GAN to estimate probability distributions in an unsupervised manner, they are well suited for unsupervised tasks such as image fusion. The GAN-based methods establish an adversarial game between the generator and the discriminator and continuously improve the performance of the fused image. Unfortunately, these methods are extremely unstable and prone to image distortion. In addition, with the development of Transformer and the mining of its powerful global feature extraction ability, scholars have begun to study whether Transformer-based methods are suitable for image fusion. However, for methods fully adapted to infrared and visible image fusion tasks, it is necessary to enhance multi-scale base information and detail textures while extracting global features, which the vast majority of algorithms combining Vision Transformer and CNN cannot do. In addition, the convolution kernel of CNN also has certain limitations for extracting image detail features and salient features.

Through the above description of the problem, we can find that most of the existing deep learning-based methods except Transformer-based one rely on the convolution operation. Although CNN has good local feature extraction ability, its small receptive field makes it difficult to model the long-range dependence, and thus, it is difficult to obtain the global features of the source images. In addition, most fusion networks introducing a pooling operation can extract rich image features using scale transformation, but it also brings a lot of information loss and artifacts.

In order to solve the above problems, the wavelet-guided pooling is applied into super feature attention (SFA) to form a novel Transformer architecture, which will be introduced in next section. The SFA mechanism can learn an efficient global representation, especially for shallow features. The number of image primitives for subsequent processing is reduced in a token space. The multi-scale features and detail information can be simultaneously extracted using wavelet-guided pooling. In the feature extraction stage, three convolutional layers are utilized to expand the channel number of the features and to enlarge the receptive field.

To be specific, the SFA attention aggregates tokens into local features by learning sparse relationships between tokens and local regions, and then, we use a self-attention machine to fully mine the local and global dependencies of the source images to capture complementary features. After that, the local features are mapped to the original space to extract global features. Finally, the deep information are extracted using embedded wavelet-guided pooling, where the multi-scale base component is transmitted to the next encoding layer and the detail components are skipped to the corresponding decoding layer after using the $l_{1}$-norm and softmax-like fusion strategy for inverse wavelet-guided pooling. As shown in Figure 1, compared with SwinFuse [29] (base on the Swin Transformer [30]), our method can extract global features while enhancing the detail textures, as shown in the green box; the reflection of the tree is clearly displayed. SwinFuse is less capable of processing multimodal images than we are, resulting in generated images darker and affecting visual senses. Our method combines the efficient global representation ability of Transformer with the detail enhancement of wavelet-guided pooling, which can extract more fine-grained features and reconstruct better-quality fused images.

The main contributions of this paper are summarized as follows:An improved Transformer architecture combined with wavelet-guided pooling and attention mechanism (super feature) is proposed, which preserves global features while enhancing detail information.A simple yet efficient fusion network is proposed, which only employs a simple $l_{1}$-norm and the softmax-like fusion strategy to achieve better fusion performance.The qualitative and quantitative experiments on several public image fusion datasets demonstrate the superiority of the proposed fusion method.

The rest of this paper is organized as follows: Section 2 discusses related work on image fusion. In Section 3, we present the details of the proposed fusion method. Section 4 shows the experimental results and compares it with some state-of-the-art methods. Finally, we present the conclusions in Section 5.

## 2. Related Work

### 2.1. Deep Learning Model for Image Fusion

Due to the high efficiency and feature representation ability of deep learning, it is widely used in the field of image fusion. In IFCNN [31], convolutional layers are first used to extract salient image features from multiple input images. Then, the convolutional features of multiple input images are fused using appropriate fusion rules. Finally, the fusion features were reconstructed using two convolutional layers to generate an information-rich fused image. The representative auto-encoder-based method is DenseFuse [24]. The encoder network composed of cascaded networks is used to extract the features of the source images; then, a manually designed fusion strategy is conducted on these features; and finally, image features are reconstructed through the decoder network. In addition, NestFuse [25] similarly adopt the above three basic steps to fuse features. In the end-to-end model termed LRRNet [32], Li et al. propose a learnable low-rank representation (LRR) approach to the fusion task, which can preserve the image details and enhance the salient features of the source images. With the rise of generative adversarial network, Ma et al. [26] first applied the GAN [33] algorithm to image fusion. However, GAN-related networks are extremely unstable, so Ma et al. proposed DDcGAN [27], which contains two discriminators to force the generator to fuse useful information from original images.

However, these methods ignore the importance of global dependencies and cannot be well enhanced for details. Therefore, our proposed method combining super feature attention and wavelet-guided pooling extracts the global information of the image while enhancing the detail texture, so as to improve the global dependence of the images.

### 2.2. Transformer for Image Fusion

The Transformer structure was proposed by Vaswani et al. [34], which was applied to machine translation. Subsequently, Dosovitskiy et al. [35] proposed the Vision Transformer (ViT) to solve the image classification problem. In recent years, many Transformer-based infrared and visible image fusion models have been proposed. The Transformer architecture also was introduced into the image fusion task [36]; the original authors proposed a two-stage training approach to extract features and developed a Transformer-based multi-scale fusion strategy that attends to both local and long-range information.

With the proposal of Swin Transformer [30], some scholars have applied it to the field of image fusion. The Swin Transformer employs window-based local attention to restrict attention to non-overlapping local windows. Despite being able to reduce the redundancy in local regions, the features of local windows still exists in shallow layers and global features cannot be fully obtained. SwinFusion [28] adopts a hybrid structure of CNN and Swin Transformer. The intra-domain fusion module extracts features of the source images through the attention mechanism and then exchanges features via the inter-domain fusion modules. The small kernel convolution layer is used to fuse the global features. Finally, the image is reconstructed with the reconstruction unit. However, it lacks long-term dependence on feature fusion, and it is not sufficient for deep global detail feature extraction. SwinFuse [29] designs a fusion network based on the residual Swin Transformer, where the fusion result is obtained through global feature extraction, fusion layer, and feature reconstruction. DATFuse [6] designs a dual-attention residual module for feature extraction and a Swin Transformer module for global complementary information preservation. However, they do not change the Swin Transformer feature extraction module, so they are also equally limited when integrating complementary features and global interactions.

Recently, THFuse [37] introduced the Transformer into the CNN-based fusion network to focus on both global and local information. In this work, they extracted the shallow features using the dual-branch CNN module, and the vision Transformer module was used to obtain the global channel and spatial relationship in the features. In TCCFusion [38], they designed the local feature extraction branch to retain local complementary information and the global feature extraction branch composed of three Transformer blocks to maintain long-range dependencies. These two modules are sufficient to capture local and global useful information. Yi et al. [39] proposed the fusion network (TCPMFNet), which is based on the auto-encoder structure and designed the Transformer–convolutional parallel mixed fusion strategy to achieve outstanding feature fusion performance. DGLT-Fusion [40] decouples global–local information learning into Transformer and CNN modules, which enables the network to extract better global–local information. TGFuse [41] was proposed as a fusion algorithm that combines Transformer and GAN. The Transformer module is simply used to learn the global fusion relationship. In contrast, we combine the Super Feature Transformer module and wavelet-guided pooling to preserve more detail texture information while learning global features. Wave-ViT [42] was proposed as a framework using wavelet transforms and self-attention learning, which is suitable for image recognition, object detection, and instance segmentation. However, that Transformer module is less capable of extracting global features than our super feature Transformer, and our method can achieve excellent results using a simple Haar wavelet pooling compared to their complex DWT-Convolution-IDWT. Our pooling operation is able to capture multi-scale features while extracting vertical, horizontal, and diagonal detail textures. Compared with our method, they only use 3×3 convolution to extract local features and do not fully extract multi-scale information and detail textures. In addition, image fusion requires more texture details of visible images, and our method can better meet the requirements of the field of image fusion, but Wave-ViT does not.

Although the above Transformer-based methods can achieve good results, they rely too much on convolutional layers when extracting local features and fail to enhance certain meaningful features. In general, the improved Vision Transformer proposed by our SFPFusion well combines the attention mechanism and Haar wavelet-guided pooling operation to extract global features while preserving texture details, which are ignored by most Vision Transformers applied in the field of image fusion.

## 3. Proposed Fusion Method

In this section, the proposed fusion method will be presented. Firstly, we give the details of the framework and model architecture. Then, the details of training phase will be given.

### 3.1. Overall Framework

As shown in Figure 2, our network framework mainly contains three main parts: feature extraction, feature fusion, and feature reconstruction. Our proposed method aims to generate a fused image by fusing local and global complementary features from source images. The following will be the details of the framework.

#### 3.1.1. Feature Extraction

**The Convolutional Block:** Given a pair of already registered testing infrared $I^{ir}\in R^{C\times H\times W}$ and visible image $I^{vi}\in R^{C\times H\times W}$ (where *C*, *H*, and *W* represent the channel number of input images, height, and width, respectively). The first stage consists of two modules, $CB_{1}$ and $WaP_{1}$. The shallow features of the images $F^{ir}$ and $F^{vi}$ are extracted with a convolutional block ($CB_{1}$), which can be expressed as follows:(1)$$\left\{F^{ir},F^{vi}\right\}=\left\{H_{CB_{1}}\left(I^{ir}\right),H_{CB_{1}}\left(I^{vi}\right)\right\}$$
where $H_{CB_{1}}\left(\cdot \right)$ represents the first convolutional block.

The convolutional block consists of multiple convolutional layers, which provides an effective way to extract local information. The $CB_{1}$ contains three convolutional layers using ReLU activation function, whose convolution kernel is 3 and stride is 1. The other main function of $CB_{1}$ is to transform the input channel number to 64, which will facilitate the subsequent extraction of deeper features by expanding the number of channel.

**Wavelet-guided Pooling:** After that, we extract multi-scale base information ($F_{LL}$) and detail features $\left(F_{VD},F_{HD},F_{DD}\right)$ using the first wavelet-guided pooling operation ($WaP_{1}$), which can be formulated as follows:(2)$$\left\{F_{LL}^{\omega },F_{VD}^{\omega },F_{HD}^{\omega },F_{DD}^{\omega }\right\}=WaP_{1}\left(F^{\omega }\right),\;\omega \in \left\{ir,vi\right\}$$

Compared with the ordinary pooling operation, wavelet-guided pooling has four kernels: (3)$$LL=\left[\begin{array}{cc} 1 & 1\\ 1 & 1 \end{array}\right]\;VD=\left[\begin{array}{cc} -1 & 1\\ -1 & 1 \end{array}\right]$$



(4)
$$HD=\left[\begin{array}{cc} 1 & 1\\ -1 & -1 \end{array}\right]DD=\left[\begin{array}{cc} 1 & -1\\ -1 & 1 \end{array}\right]$$



The multi-scale component LL obtains the global multi-scale base information, while the detail components VD, HD, and DD extract the vertical, horizontal, and diagonal detail textures.

The traditional pooling operation has some limitations. For max-pooling, significant features are lost in the pooling operation and images cannot be reconstructed accurately. Average pooling, on the other hand, dilutes meaningful features. Inspired by [43], we use Haar wavelet-guided pooling in the Super Feature Pooling Transformer module. Haar wavelet is one of the classic wavelet transforms. Compared with the traditional pooling operation, Haar wavelet-guided pooling decomposes the original image into different component channels, so as to extract multi-scale information and texture details. Moreover, it has minimal information loss and can reconstruct images completely without any post-processing step.

For our framework, only the multi-scale base information $F_{LL}$ passes to the encoding layer and the detail features $\left(F_{VD},F_{HD},F_{DD}\right)$ are skipped to the corresponding unpooling layer after feature fusion. Our unpooling operation is based on the inverse transform of wavelet-guided pooling, which can reconstruct the image accurately and minimize the information loss.

**Super Feature Pooling Transformer:** However, in [43], they only use convolutional layers to extract image features, which has certain limitations. Given that the inherent small receptive field of CNN cannot effectively extract global information of source image, the Super Token Transformer [44] is introduced. It can obtain efficient global representation and enhance detail information simultaneously. Specifically, we improve it and embed wavelet-guided pooling into the Transformer architecture, termed as Super Feature Pooling Transformer, so that the global features can be augmented to obtain richer features (multi-scale features and detail information).

To be specific, our Super Feature Pooling Transformer is mainly composed of four modules: position encoding (PE), super feature attention (SFA), embedded pooling layer (EPL), and patch merging module (PM).

As shown in Figure 2, the multi-scale features $F_{LL}^{1}$ extracted by $WaP_{1}$ are used as the input of the Super Feature Pooling Transformer. For convenience of expression, we replace $F_{LL}^{1}$ with $F_{in}$.

Given the input features $F_{in}\in R^{C\times H\times W}$, we firstly add position information to all the tokens by using position encoding (PE). PE containing a 3×3 depth-wise convolution can learn the local representation better than absolute position encoding (APE) [34] and relative position encoding (RPE) [30,45]. The formula is defined as follows:(5)$$T=H_{pos}\left(F_{in}\right)$$
where $H_{pos}\left(\cdot \right)$ represents the positional encoding, which is used to add position information into all the tokens. $T\in R^{C\times H\times W}$ are the visual tokens we obtain.

After extracting the position information, we next employ super feature attention to extract global features:(6)$$F_{SF}=SFA\left(LN\left(T\right)\right)+T$$
where SFA(·) represents the super feature attention mechanism and LN(·) denotes LayerNorm operation. In the following, we will describe the process of super feature attention in detail.

Given the visual tokens $T\in R^{C\times H\times W}$, we first compute the initial super features $S\in R^{C\times m\times n}$ using average pooling in regular grid regions, where the grid size is h×w, $m=\frac{H}{h}$, and $n=\frac{W}{w}$. The number of super features is m×n. In order to better extract complementary features and global features, it is necessary to calculate the mapping associations $Q\in R^{mn\times hw\times 9}$ between *T* and *S*. The details of SFA operation, please refer to our Appendix A.

Following the super feature attention module, an embedded pooling layer is deployed to augment multi-scale features and detail information, which can be defined as follows:(7)$$\left\{F_{LL},F_{VD},F_{HD},F_{DD}\right\}=\mathrm{Pooling}\left(F_{SF}\right)$$
where Pooling(·) represents the embedded wavelet-guided pooling operation.

Our embedded pooling layer has the same structure as the above wavelet-guided pooling, as shown in Figure 2. The extracted multi-scale base information is passed to the next layer, and the detail features are given in the unpooling operation for image reconstruction.

As shown in Figure 3a, we can observe that the super feature attention mechanism well builds long-range dependencies and extracts the global salient features. In addition, the multi-scale base information (b) and detail information (c)–(e) obtained by the embedded pooling layer are well enhanced, which helps to reconstruct the image with fine features.

Finally, the extracted multi-scale base features $F_{LL}$ go through the patch merging module to increase the number of output channels. With super feature attention, we can make full use of long-range dependencies to extract enhanced global features $F_{GF}$.
(8)$$F_{GF}=H_{PM}\left(F_{LL}\right)$$
where $H_{PM}\left(\cdot \right)$ represents the patch merging module.

In summary, with the combination of PE, SFA, and EPL, the fusion network is able to fully implement local information extraction and global information integration.

After extracting enhanced global information and detail textures from $SFP_{1}$ in the second stage, we put the output features into the third stage $SFP_{2}$ with the same structure to extract deeper features again, as shown in Figure 2. In the fourth stage, when the global features and complementary features of the three stages are extracted, they are passed to the convolution block ($CB_{2}$) to obtain local semantic information and are mapped into the high-order feature space. The network architecture of the specific feature extraction stages is shown in Table 1.

#### 3.1.2. Fusion Strategy

**Global Features Fusion Strategy:** It is well known that choosing an appropriate fusion strategy is very important for image fusion. In this work, we use the fusion strategy based on $l_{1}$-norm and softmax-like. The $l_{1}$-norm represents the sum of the absolute values of the elements in the vector.

Given the extracted infrared $F_{out}^{ir}\in R^{C\times H\times W}$ and visible $F_{out}^{vi}\in R^{C\times H\times W}$ features, we first sum the vectors in each channel using the $l_{1}$-norm operation and then the weighting maps are generated using the softmax-like operation, which can be calculated with the following:(9)$$W_{\omega }\left(x,y\right)=\frac{∥F_{out}^{\omega }{\left(x,y\right)∥}_{1}}{\sum_{i\in \left\{ir,vi\right\}}{∥F_{out}^{i}\left(x,y\right)∥}_{1}},\;\omega \in \left\{ir,vi\right\}$$
where ${∥\cdot ∥}_{1}$ denotes the $l_{1}$-norm. (x,y) indicates the corresponding position in deep features ($F_{out}^{ir}$ and $F_{out}^{vi}$) and weighting maps ($W_{ir}$ and $W_{vi}$ ).

Finally, the extracted features in each channel are multiplied by the weighting maps to obtain the fused feature maps $F_{f}$. The specific process of the fusion strategy is shown in Algorithm 1.

**Detail Features Fusion Strategy:** For the pooling layer, the detail information preserved by wavelet-guided pooling are sparse but also very significant. Therefore, it can achieve good results without complex operations. Moreover, it is very important to preserve as much texture details as possible in the field of image fusion. In this work, we sum the saved infrared and visible detail information corresponding to the scale, which can be formulated as follows:(10)$$F_{det}^{f}=F_{det}^{ir}+F_{det}^{vi},\;det\in \left\{VD,HD,DD\right\}$$
where $F_{det}^{ir}$ and $F_{det}^{vi}$ represent the texture details of infrared and visible images in the vertical, horizontal, and diagonal directions, respectively. $F_{det}^{f}$ represents the fused features of the corresponding scale detail parts.
**Algorithm 1** Procedure for fusion strategy. 
**Require:** 
     the extracted infrared $F_{out}^{ir}\in R^{C\times H\times W}$ and visible $F_{out}^{vi}\in R^{C\times H\times W}$ features 
**Ensure: **
     fused feature maps $F_{f}$;   1:Take the absolute value of each pixel value;   2:Make the sum of pixel values within each channel;   3:Calculate the weighting maps by softmax-like operation according to Equation (Equation 9);   4:**for** each i∈[1,C]
**do**   5:   Multiplying features within *i*-th channel and weighting maps, which can be expressed as follows:
(11)$$F_{f}^{i}=\sum_{\omega \in \left\{ir,vi\right\}}W_{\omega }\left(x,y\right)\times F_{out-i}^{\omega }\left(x,y\right)$$   6:**end for**   7:Concatenate the channel of $F_{f}^{i}$, and the number of channel is restored to *C*;   8:**return** $F_{f}$;

#### 3.1.3. Feature Reconstruction

In this phase, the fused features are used to reconstruct the image through four convolutional blocks and three wavelet-guided unpoolings. Specifically, our wavelet-guided unpooling is the inverse operation of wavelet-guided pooling. The detail information extracted in the feature extractor adopts an addition strategy to the corresponding scale and is used to reconstruct the image together with the fused global feature, as shown in Figure 2.

In addition, our image reconstruction stage can fully recover the image features without any post-processing operations, so as to achieve the effect of minimum information loss. The specific details of the image reconstructor are shown in Table 2.

### 3.2. Training Phase

Our fusion network is based on an auto-encoder and the fusion strategy needs to be discarded during the training phase. In the training phase, the trained feature extractor can better extract global features and detail information about source images, and the trained feature reconstructor can recover image features well and reduce information loss, as shown in Figure 4.

In order to reconstruct the input image more accurately, the loss function is composed of a pixel loss function $L_{pixel}$ and a structural similarity loss function $L_{ssim}$, which can be formulated as follows:(12)$$L_{toal}=L_{pixel}+\lambda L_{ssim}$$
where λ denotes the tradeoff value between $L_{pixel}$ and $L_{ssim}$.

Specifically, $L_{pixel}$ is mean square error, which is used to make sure that the reconstructed image is more similar to the input image at the pixel level, as calculated in the following equation:(13)$$L_{pixel}={∥O-I∥}_{F}^{2}$$
where *O* and *I* indicate the output and input images, respectively. ${∥\cdot ∥}_{F}^{2}$ is the $l_{2}$-norm.

In addition, the SSIM loss $L_{ssim}$ is formulated as follows:(14)$$L_{ssim}=1-SSIM\left(O,I\right)$$
where SSIM(·) denotes the structural similarity between the output image and input image [46].

## 4. Experimental Results and Analysis

### 4.1. Experimental Settings

In the training phase, we randomly selected 40,000 images from MS-COCO [47] to train our auto-encoder network, where all images are resized to 224×224. The batch size, epochs, and learning rate are set to 4, 4, and $1\times 10^{-4}$, respectively. The hyperparameter λ in Equation (Equation 12) is set as 10, and we use the same tradeoff value in all experiments. Moreover, our iterations are i=1. All the involved experiments are conducted on an NVIDIA RTX 3090Ti GPU (NVIDIA, Santa Clara, CA, USA) and Intel Core i7-10700 CPU (Intel, Santa Clara, CA, USA). In the test phase, for RGB images, we first convert the visible image to YCbCr color space, and then, take the Y channel of the visible images into our proposed network together with the infrared images for fusion. Finally, the fused image is converted back to the RGB color space by concatenating the Cb and Cr channels of the visible images.

To comprehensively evaluate the proposed method, we perform qualitative and quantitative experiments on the MSRS dataset [7] with 361 image pairs, the LLVIP dataset [48] with randomly selected 389 image pairs, and the TNO dataset [49] with randomly selected 16 image pairs. We compare our method with eight state-of-the-art (SOTA) approaches, including DenseFuse [24], FusionGAN [26], SwinFuse [29], U2Fusion [50], AUIF [51], CUFD [23], MUFusion [52], and AEFusion [53]. The implementations of these approaches are publicly available and the initial parameters of the compared methods remain the same.

Five quality metrics are selected for the fair and quantitative comparison between our fusion method and other SOTA methods, including standard deviation (SD) [54], visual information fidelity (VIF) [55], average gradient (AG) [56], entropy (EN) [57], and $Q_{abf}$ [58]. SD reflects the distribution and contrast of the fused image, which is consistent with the visual perception. VIF is the metric based on natural scene statistics and the conception of image information extracted by the human visual system. AG quantifies the gradient information of the fused image and represents its detail textures. EN is used to represent the amount of information contained in image. $Q_{abf}$ measures the amount of edge information. Moreover, a fusion algorithm with larger SD, VIF, AG, EN, and $Q_{abf}$ indicates better fusion performance.

### 4.2. Comparative Experiments

In this section, our method will be compared with eight SOTA methods both qualitatively and quantitatively to show the superiority of our algorithm on MSRS, LLVIP, and TNO datasets. For more experiments, please refer to our Appendix A.

#### 4.2.1. Fusion Results on MSRS Dataset

One set of source image pairs of the MSRS dataset and their corresponding fused images obtained using different methods are demonstrated in Figure 5. In daytime scenarios, thermal radiation information from infrared images should be used as complementary information to visible images. As a whole, although FusionGAN enhances the thermal radiation information, it causes serious spectral contamination and affects the overall visual sense. SwinFuse and AUIF extract too much background features of infrared images, making the whole scene submerged in darkness, which are completely not suitable for daytime scenes. Moreover, as shown in the green and red boxes, DenseFuse and U2Fusion weaken infrared targets and also fail to retain detail information of visible images. Although AEFusion preserves more texture details, it similarly weakens salient features. In addtion, CUFD and MUFusion are able to synthesize the texture details of visible images and the salient features of infrared images. Unfortunately, a large number of artifacts are introduced into fused results, resulting in poor visual perception. Only our method enhances the thermal radiation information while preserving the texture information and provides a pleasing visual effect.

In order to better evaluate the quality of the generated images, we conduct qualitative comparisons on 361 image pairs from MSRS datasets to verify the effectiveness of our algorithm, as shown in Table 3. It is worth noting that our method exhibits excellent superiority in all five metrics. The best SD metric indicates that our fused images have richer contrast information, which achieves a good visual effect. The best result in VIF metric indicates that our results have better visual perceptual performance, which is consistent with human visual perceptions. For the AG and $Q_{abf}$ metrics, they achieve the best results, meaning that our results have richer texture details and more edge information, benefiting from our proposed wavelet-guided pooling. Moreover, the best EN metric shows that our fused images contains more scene information.

#### 4.2.2. Fusion Results on LLVIP Dataset

We further randomly select 389 image pairs of nighttime scenes on the LLVIP dataset to demonstrate the effectiveness of our method. In the dark scenes, the infrared images contain a lot of thermal radiation information and detail textures, which will complement the visible images containing limited detail information. As shown in Figure 6, FusionGAN fails to preserve texture details and the background suffers from severe contamination, causing visual conflicts. For SwinFuse and AUIF, the infrared features are weakened. These too dark images lead to the retention of little detail information, making people obtain less useful information. As shown in the red box, U2Fusion and DenseFuse do not retain more texture details, mainly due to inadequate feature extraction of infrared images. Although CUFD and AEFusion retain more thermal radiation information, they blur edge information and produce extensive artifacts at the same time, resulting in overall image blur. In addition, MUFusion introduces additional information into the fused images in some cases, which can be seen from the ground in a green box. It is worth noting that, as shown in the green box, our algorithm can very clearly show the reflection of branches in the dark, which has a great advantage compared with the algorithms that blur the targets.

The quantitative results of five metrics on image pairs from the LLVIP dataset are presented in Table 4. It can be observed that our method ranks first in three metrics. Our method achieves the highest AG metric, indicating that our fused images possess more detailed textures. Moreover, the best metric $Q_{abf}$ implies that more edge information is preserved in fused results. For the best EN metric, it demonstrates that our method has good superiority in containing valid information. In addition, our method still has a pleasing performance on the VIF metric, only following FusionGAN. It shows that our fused images can still achieve good visual effects. On the SD metric, the proposed method follows FusionGAN and AEFusion by a narrow margin. And, with the help of the visualization results, we can see that FusionGAN and AEFusion blur a lot of detail information.

#### 4.2.3. Fusion Results on TNO Dataset

For grayscale images of infrared and visible image fusion, we select 16 image pairs on the TNO dataset for comparison. As shown in Figure 7, DenseFuse, U2Fusion, and AEFusion weaken the thermal radiation information, and the part of sky introduces plenty of artifacts. Moreover, DenseFuse and AEFusion also blur the edge information, which can be seen from red box. For MUFusion, it introduces too much extra information, causing conflicts in human perception. FusionGAN retains too much of the infrared features, causing loss of details and background contamination. In contrast, SwinFuse, AUIF, and CUFD preserve the salient features well and can achieve good results. However, it also causes the loss of details. As shown in the red box, they all do not preserve the detail information of branches and fences well. In general, our method preserves the salient features and texture details well, which is attributed to our proposed method of wavelet-guided pooling combined with super feature attention.

The comparative subjective results of different methods on the TNO dataset are shown in Table 5. Our method achieves the best results on AG and $Q_{abf}$ metrics, which indicates that our fused results contain richer gradient textures and more edge information. On the VIF metric, our method has a small gap compared with CUFD, indicating that our method can still achieve good visual results. However, in terms of SD and EN metrics, our method does not achieve as good results as the other two datasets. This is justified. Compared with the previous two datasets (all RGB images), TNO as the dataset of grayscale images mainly contains salient features. The visible images in TNO dataset do not contain more obvious texture details as visible images in RGB. From our fusion network, it can be seen that our method combines wavelet-guided pooling and super feature attention, which prefers to preserve the intensity information and global features of the source images. Therefore, it is understandable that our method performs worse on the TNO dataset than the first two datasets. Even so, the results of the two metrics can still be kept within the top three.

In general, the visualization results show that our method has obvious advantages in preserving global features and maintaining detail information, while achieving satisfied visual effects. The objective evaluation metrics also confirm that our method can achieve better fusion performance.

### 4.3. Comparison of the Efficiency

We also conduct comparison experiments about the average running time of different fusion methods. We use 361 pairs of infrared and visible images with the same size (640×480) to test the operational efficiency of these methods. As shown in Table 6, U2Fusion achieves relatively better performance on efficiency comparison. For our SFPFusion, due to the improvement in the Transformer and combined with wavelet pooling operation guidance, this is somewhat slower than methods with fewer parameters. However, our method is able to achieve better results than SwinFuse, which is based on Swin Transformer. In addition, our SFPFusion still obtains a comparable result on the efficiency, with an average time of less than 0.1 s per image.

### 4.4. Ablation Studies

#### 4.4.1. Analysis of Transformer

Since our network introduces the super feature attention mechanism combined with wavelet-guided pooling, making it better adapted to infrared and visible image fusion, we use the original Super Token Transformer [44] in ablation experiments to demonstrate the advantages of our embedded wavelet-guided pooling. As shown in Figure 8d, although the Super Token Transformer without pooling preserves the features of visible images better, it weakens the salient features from infrared images and the detail part is not enhanced compared with our method. Moreover, the sky also forms an over-exposed scene. In addition, it can also be seen from the evaluation metrics (Table 7) that our fusion results are better than the original framework without pooling as a whole.

In addition, we conduct Vision Transformer ablation experiments to confirm that our improved Transformer is better than the original ViT. We introduce the original ViT [35] structure, which does not have embedded pooling, and thus, the unpooling operations of $WaUP_{2}$ and $WaUP_{3}$ are not included in the framework of the ablation experiments. As shown in Figure 8e, compared with it, our fusion result can preserve more sufficient texture details and global features and can accurately reconstruct the image without introducing noise. The quantitative evaluation in Table 7 also shows that the quality of our generated images are all better than using ViT.

#### 4.4.2. Analysis of CNN

As we stated in Section 1, the size of convolution kernel of CNN determines how many features it can extract from its receptive field, which is its limitation. Therefore, we conduct ablation experiments combining CNN with pooling to confirm that our super feature attention mechanism can better capture long-range dependencies, perform effectively global representation, and obtain global features of the source images. When training the auto-encoder and the hyperparameter λ in Equation (Equation 12) is set as 10, the auto-encoder cannot reconstruct the image. In the ablation experiment, the hyperparameter λ is set to 100 to reconstruct the image well. As shown in Figure 8f, the fusion results using CNN weaken the salient features and fail to model the long-range dependence of the image, resulting in a very poor visual perception. Moreover, it also introduces a lot of artifacts in the sky. In contrast, our method is superior to the fusion results using CNN in both visual results and quantitative metrics (Table 7).

#### 4.4.3. Analysis of Structure

In our framework, after the first convolutional block ($CB_{1}$), we utilize wavelet-guided pooling to extract the first multi-scale features and detail information. In the ablation experiments, we discarded the first layer pooling ($WaP_{1}$). As shown in Figure 8g, artifacts still appear in the sky despite sufficient features extracted from the generated image. And, as shown in Table 7, the evaluation metrics are all significantly lower than our method. This also indicates that $WaP_{1}$ is also important in extracting the information of the first layer.

In addition, we perform ablation experiments on the number of Super Feature Pooling Transformers. We use a SFP module and the visualization result is shown in Figure 8h. Although it preserves good salient features, the enhancement of the detail part is not as good as our two SFP modules, and the sky also forms overexposure. The superiority of our use of two SFP modules can also be seen in Table 7. Moreover, we try to use three SFP modules to extract features but unfortunately fail when training the auto-encoder. The reason for the failure is that the last layer of the reconstructed network uses a Sigmoid activation function, which causes the gradient to disappear in the network with deep superposition.

#### 4.4.4. Analysis of Global Features Fusion Strategy

For the global feature fusion strategy, we use the simple $l_{1}$-norm and softmax-like function-based strategy, which can also achieve good fusion results. Thus, in the ablation experiments, the addition strategy and the average strategy are performed. As shown in Figure 8i,j, the addition fusion strategy excessively enhances the salient features, resulting in over-exposure of the image. However, using the average strategy weakens the infrared intensity, resulting in the overall dark image, and also introduces a lot of noise in the sky. In addition, from the evaluation of metrics (Table 7), it can also be found that our method is generally better than the other two strategies, which confirms that our fusion strategy is simple and efficient.

#### 4.4.5. Analysis of Detail Features Fusion Strategy

As mentioned in Section 3.1.2, the extracted detail information is sparse and clear, which can lead to better fusion performance without complicated operations. As shown in Figure 3 and Figure 9, the generated feature maps ($F_{VD}^{2}$, $F_{HD}^{2}$, and $F_{DD}^{2}$) clearly show the detail textures. The detail features extracted with our method are very prominent, so they can be fused well using only a simple addition strategy. In the ablation experiments, we use the average strategy for the fusion strategy of detailed features. As shown in Figure 8k, using the average strategy weakens the detail intensity, which results in the overall blurring of the images. Moreover, it can be seen from the quantitative results (Table 7) that all the metrics are significantly lower than those of our method. Therefore, the addition fusion strategy is very effective for detail feature fusion.

#### 4.4.6. Analysis of Different λ

In Equation (Equation 12), our hyperparameter λ is set to 10 and the same value is used in all experiments. Therefore, it is necessary to compare the metrics of fused images with different λ. As shown in Table 8, the best experimental results can be achieved when λ = 10.

## 5. Conclusions

In this work, we propose a novel Vision Transformer-based fusion network (SFPFusion) which combines wavelet-guided pooling and the super feature attention mechanism to obtain effective global features while enhancing detail information. Specifically, the super feature attention mechanism decomposes ordinary global attention into the product of sparse correlation maps and low-dimensional attention, which can capture more long-range dependencies. Moreover, combined with the embedded wavelet-guided pooling layer, multi-scale base information and detail textures are well enhanced after extracting global features. We pass the preserved detail features (vertical, horizontal, and diagonal detail textures) to the unpooling layer, which can accurately recover the structure and texture information of the images. In addition, the proposed fusion network is also simple yet efficient by adopting the simple $l_{1}$-norm and softmax-like fusion strategy. The experiments on three public available datasets demonstrate the effectiveness of our proposed method in terms of qualitative results and quantitative evaluation. Ablation experiments also confirm the role of different components in our fusion network. In the future, texture and color information will play an important role in image content recognition. As a result, infrared and visible image fusion can be widely used in the pretreatment of the image retrieval system [59], etc.

## Figures and Tables

**Figure 1 sensors-23-07870-f001:**
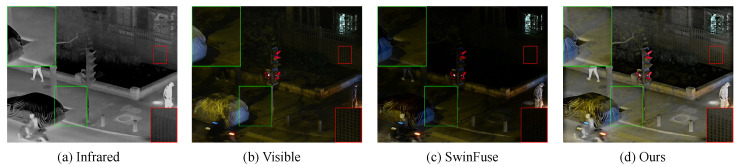
Visualization results of different Transformer-based methods. From left to right: infrared image, visible image, Swin Transformer-based fusion model (SwinFuse), and the proposed SFPFusion.

**Figure 2 sensors-23-07870-f002:**
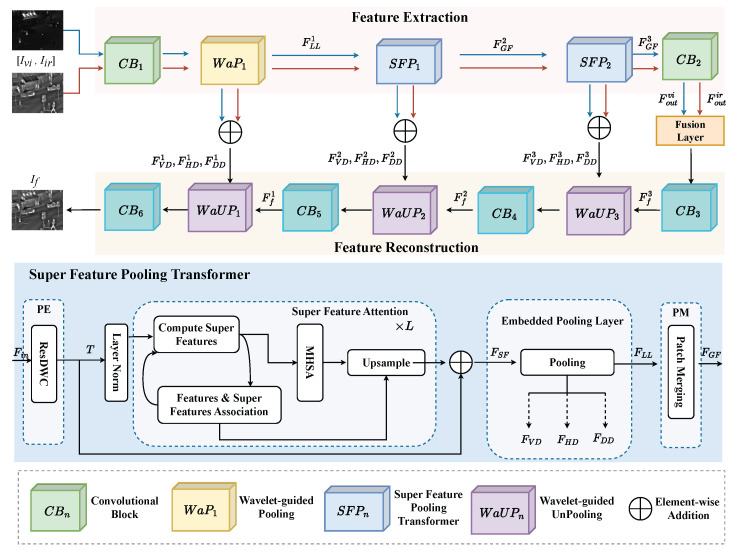
The proposed module combining super feature attention and wavelet-guided pooling. *L* represents the number of times attention extracts features. The features $F_{SF}$ extracted with super feature attention are passed to the embedded pooling layer to extract deep multi-scale features ($F_{LL}$) and detail textures ($F_{VD}$, $F_{HD}$, and $F_{DD}$). In the reconstruction, the components are aggregated by the wavelet-guided unpooling.

**Figure 3 sensors-23-07870-f003:**
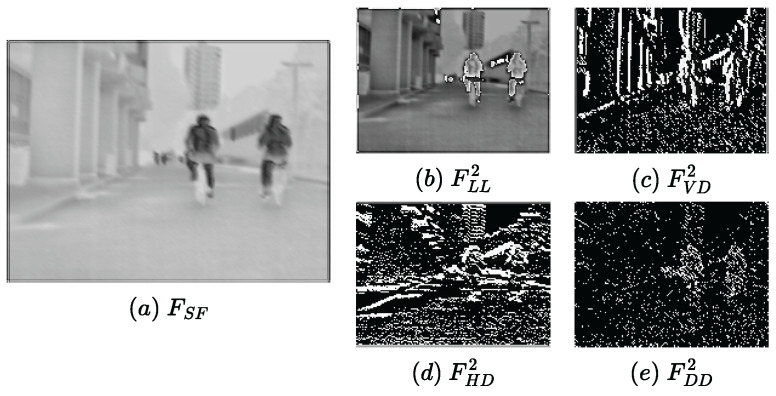
Visualization results of feature maps within the $SFP_{1}$. $F_{SF}$ represents the global features extracted by the attention mechanism. Then, the global multi-scale information is obtained by the multi-scale kernels LL, VD, HD, and DD to capture vertical, horizontal, and diagonal texture details.

**Figure 4 sensors-23-07870-f004:**
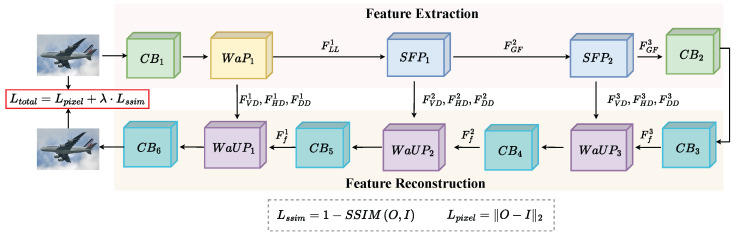
The framework of training process. In this process, we train an auto-encoder network without fusion strategy.

**Figure 5 sensors-23-07870-f005:**
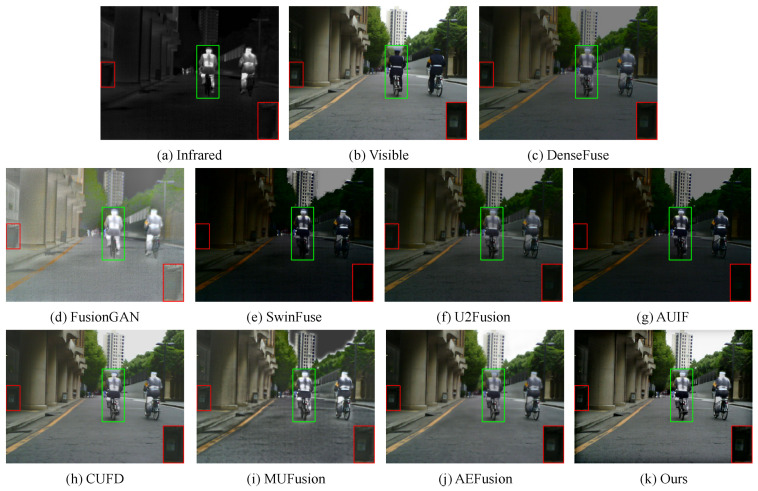
Qualitative comparison of our method with 8 state-of-the-art methods on the MSRS dataset.

**Figure 6 sensors-23-07870-f006:**
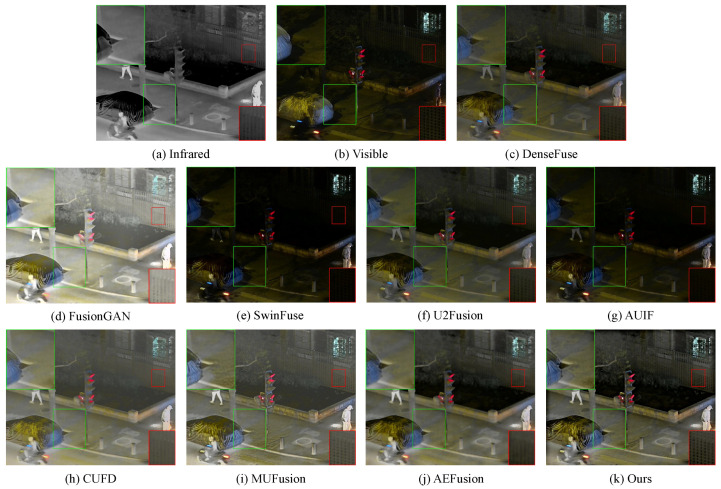
Qualitative comparison of our method with 8 state-of-the-art methods on the LLVIP dataset.

**Figure 7 sensors-23-07870-f007:**
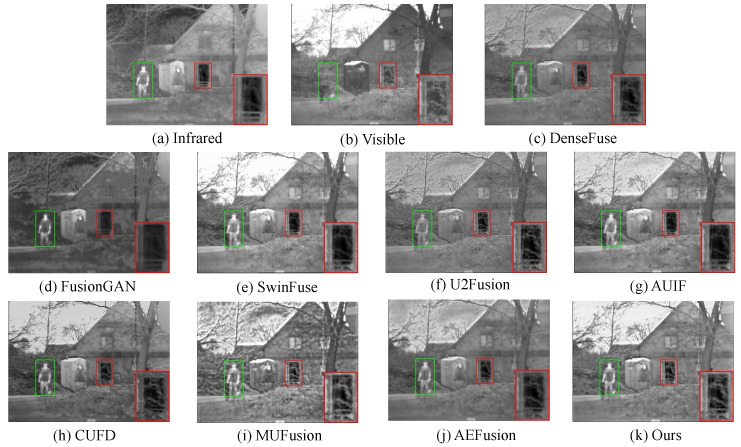
Qualitative comparison of our method with 8 state-of-the-art methods on the TNO dataset.

**Figure 8 sensors-23-07870-f008:**
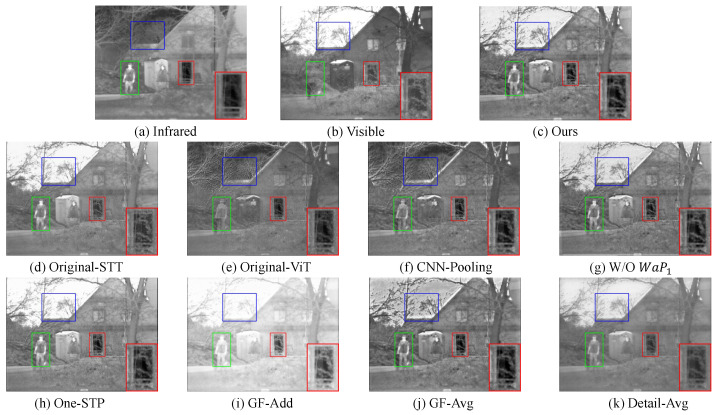
Visualized results of ablation studies. From (**c**) to (**k**): fused results of our method, fused results of using original Super Token Transformer, fused results of using original Vision Transformer, fused results of using CNN with pooling, without $WaP_{1}$, global features using average strategy, global features using addition strategy and detail features with different directions using average strategy.

**Figure 9 sensors-23-07870-f009:**
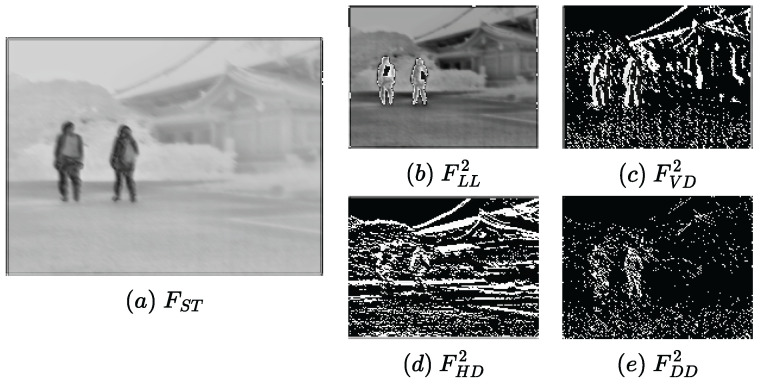
Visualization results of feature maps within the $SFP_{1}$.

**Table 1 sensors-23-07870-t001:** Network architecture of feature extraction. **Input** and **Output** denote the number of channels in the corresponding feature maps.

	Stage	Block	Layer Name	Kernel	Input	Output	Size	Activation
Extraction	Stage1	${\mathrm{CB}}_{1}$	Layer1	1×1	1	3	224×224	ReLU
			Layer2	3×3	3	64	224×224	ReLU
			Layer3	3×3	64	64	224×224	ReLU
		${\mathrm{WaP}}_{1}$	Pooling	2×2	64	64	112×112	
	Stage2	${\mathrm{SFP}}_{1}$	CPE	$\left[\begin{array}{c} 3\times 3,\;64,\;\;64\\ \mathrm{grid}\;8,\;\mathrm{heads}\;1\\ 2\times 2,\;64,\;\;64\\ 3\times 3,\;64,\;\;128 \end{array}\right]\times 3$			112×112	
			SFA				112×112	
			Pooling				56×56	
			PM				56×56	
	Stage3	${\mathrm{SFP}}_{2}$	CPE	$\left[\begin{array}{c} 3\times 3,\;128,\;\;128\\ \mathrm{grid}\;4,\;\mathrm{heads}\;2\\ 2\times 2,\;128,\;\;128\\ 3\times 3,\;128,\;\;320 \end{array}\right]\times 5$			56×56	
			SFA				56×56	
			Pooling				28×28	
			PM				28×28	
	Stage4	${\mathrm{CB}}_{2}$	Layer1	3×3	320	512	28×28	ReLU

**Table 2 sensors-23-07870-t002:** Network architecture of feature reconstruction. **Input** and **Output** denote the number of channels in the corresponding feature maps.

	Blocks	Layers	Kernel	Input	Output	Activation
Reconstruction	${\mathrm{CB}}_{3}$	Layer1	3×3	512	256	ReLU
	${\mathrm{CB}}_{4}$	Layer1	3×3	256	256	ReLU
		Layer2	3×3	256	256	ReLU
		Layer3	3×3	256	256	ReLU
		Layer4	3×3	256	128	ReLU
	${\mathrm{CB}}_{5}$	Layer1	3×3	128	128	ReLU
		Layer2	3×3	128	64	ReLU
	${\mathrm{CB}}_{6}$	Layer1	3×3	64	64	ReLU
		Layer2	3×3	64	1	Sigmoid

**Table 3 sensors-23-07870-t003:** Quantitative results on 361 image pairs from the MSRS dataset. (**Bold**: best, red: second best, blue: third best).

Methods	SD	VIF	AG	$Q_{\mathrm{abf}}$	EN
DenseFuse [24]	7.4237	0.6999	2.0873	0.3572	5.9340
FusionGAN [26]	7.1758	0.8692	3.1193	0.2110	5.9937
SwinFuse [29]	4.9246	0.4102	1.9673	0.1720	4.4521
U2Fusion [50]	6.8217	0.5863	2.0694	0.2972	5.5515
AUIF [51]	5.2622	0.3981	1.8238	0.1970	4.6460
CUFD [23]	7.6384	0.6488	2.9003	0.4397	6.0652
MUFusion [52]	6.9233	0.6086	3.1474	0.4110	5.9682
AEFusion [53]	8.2104	0.8548	2.6968	0.4239	6.5374
Ours	**8.2650**	**0.9214**	**3.8950**	**0.5361**	**6.5636**

**Table 4 sensors-23-07870-t004:** Quantitative results on 389 image pairs from the LLVIP dataset. (**Bold**: best, red: second best, blue: third best).

Methods	SD	VIF	AG	$Q_{\mathrm{abf}}$	EN
DenseFuse	9.2963	0.7503	2.6714	0.3458	6.8287
FusionGAN	**10.0823**	**1.0263**	2.1706	0.2765	7.1741
SwinFuse	7.5469	0.6290	2.9580	0.3392	6.0825
U2Fusion	9.4256	0.7212	2.3685	0.3425	6.7588
AUIF	7.5433	0.5877	2.8790	0.3348	6.1555
CUFD	9.1701	0.7187	2.5198	0.3264	6.8448
MUFusion	8.7452	0.7875	3.5412	0.4128	6.9242
AEFusion	9.8400	0.6302	2.0397	0.1477	7.2764
Ours	9.8103	0.9671	**4.7353**	**0.6088**	**7.3006**

**Table 5 sensors-23-07870-t005:** Quantitative results on 16 image pairs from the TNO dataset. (**Bold**: best, red: second best, blue: third best).

Methods	SD	VIF	AG	$Q_{\mathrm{abf}}$	EN
DenseFuse	9.2203	0.7349	3.8804	0.4465	6.8256
FusionGAN	8.1234	0.6197	2.8120	0.2260	6.4629
SwinFuse	9.2633	0.7982	5.5986	0.4542	6.9484
U2Fusion	9.3869	0.7200	5.4456	0.4653	6.9395
AUIF	9.2805	0.7482	5.2820	0.4513	7.0402
CUFD	9.4136	**0.8781**	4.5178	0.3782	7.0743
MUFusion	**9.5379**	0.7851	5.5756	0.3818	**7.3032**
AEFusion	9.4655	0.7803	3.4114	0.3074	7.0716
Ours	9.4400	0.8021	**6.2300**	**0.4802**	7.1578

**Table 6 sensors-23-07870-t006:** The average inference time (unit: second) on 361 pairs of images from MSRS dataset. (**Bold**: best).

Method	DenseFuse	FusionGAN	SwinFuse	U2Fusion	AUIF	CUFD	MUFusion	AEFusion	Ours
Inference time	0.1813	0.0677	0.2592	**0.0342**	0.1158	72.6157	0.7045	0.2244	0.0739

**Table 7 sensors-23-07870-t007:** The average values of the five objective metrics obtained with different ablation studies on TNO dataset. (**Bold**: best, red: second best).

Strategies	SD	VIF	AG	$Q_{\mathrm{abf}}$	EN
Ours	9.4400	0.8021	6.2300	**0.4802**	**7.1578**
Original-SFT	9.4101	**0.8451**	6.1572	0.4696	7.0922
Original-ViT	8.9065	0.6971	5.6691	0.4040	6.6331
CNN-Pooling	8.9091	0.7100	5.8372	0.4652	6.7445
W/O $WaP_{1}$	9.3070	0.7713	5.9607	0.4712	6.9916
One-SFP	9.3145	0.7935	6.0023	0.4697	7.0172
GF-Add	**10.1152**	0.7116	4.4128	0.3714	6.7826
GF-Avg	9.1915	0.7884	**6.5182**	0.4772	7.0576
Detail-Avg	9.1556	0.6812	3.3329	0.3809	6.9184

**Table 8 sensors-23-07870-t008:** The average values of the five objective metrics obtained with different λ on TNO dataset. (**Bold**: best, red: second best).

λ	SD	VIF	AG	$Q_{\mathrm{abf}}$	EN
Ours (λ = 10)	9.4400	**0.8021**	**6.2300**	0.4802	7.1578
λ = 1	**9.5901**	0.7964	6.2220	0.4768	**7.2029**
λ = 100	9.4001	0.7999	6.1762	**0.4820**	7.1582
λ = 1000	9.4644	0.7907	6.0682	0.4800	7.1623
λ = 10,000	9.5537	0.8010	6.1070	0.4763	7.1716

## Data Availability

The implementation of this work can be available at https://github.com/draymondbiao/SFPFusion.

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
