# Peer review of "SFPFusion: An Improved Vision Transformer Combining Super Feature Attention and Wavelet-Guided Pooling for Infrared and Visible Images Fusion"

_sensors, 2023, doi:10.3390/s23187870_

Round 1

Reviewer 1 Report

The main proposed approach has novelty in contribution and methodology. Revision in terms of technical details is needed before publication. Also, paper organization can be improved. In this respect, some comments are suggested to describe technical details.

1. How do you select the tradeoff value between Lpixel and Lssim in the Eq. 12? Did you use a same tradeoff value in all experiments?

2. Did you implement all of the compared methods in the Table 3? If no, add related references in the Table. If yes, discuss about the initial parameters of compared methods.  

3. In some cases, fusion should be performed in real time. So, it is suggested to discuss about the runtime of your proposed approach briefly.

4. What is the meaning of the sentence “As mentioned in Section 3.1.2 , the extracted detail information is sparse and the 480 effect is significant” in the section 4.3.5? Discuss in a clear way.

5. Your proposed approach can be used widely in image retrieval systems as preprocess. For example, I find a paper titled “Innovative local texture descriptor in joint of human-based color features for content-based image retrieval”, which has enough relation. Cite this paper and discuss about potential future works briefly.   

6. Add some image examples of the fusion output images. 

7. Which kind of pooling did you use in Super Feature Pooling Transformer module? and Why?

Review the whole text in terms of possible English grammar mistakes or typing errors. 

Reviewer 2 Report

The authors have proposed a novel vision transformer-based fusion network that combines wavelet-guided pooling and super feature attention mechanism to obtain effective global features while enhancing detailed information. The topic of the manuscript is interesting and well organized. However, similar works have been published in the field in recent years, which the authors have not mentioned. Novelty and contribution are not clear compared to them. Also, manuscript writing is evaluated at a low level. So, I can't recommend accepting it.

1)     The manuscript does not mention the following completely related works. The novelty of this manuscript is questionable.

-        D. Rao, T. Xu, and X.-J. Wu, "Tgfuse: An infrared and visible image fusion approach based on transformer and generative adversarial network," IEEE Transactions on Image Processing, 2023.

-        T. Yao, Y. Pan, Y. Li, C.-W. Ngo, and T. Mei, "Wave-vit: Unifying wavelet and transformers for visual representation learning," in European Conference on Computer Vision, 2022: Springer, pp. 328-345.

2)     Please avoid using terms like new and novel in the title.

3)     Provide references for metrics used.

4)     The manuscript has several writing errors. I have mentioned only some of them in the abstract below. The wording throughout the manuscript must benefit from copy-editing by a professional editor.

-        Infrared  The infrared

-        which preserves  that preserves

-        networks(CNN)  networks (CNNs)

-        obtains  obtain

-        to extract  for extracting

-        Transformer based  transformer-based

-        task and other down-stream  tasks and other downstream

-        into fusion  to the fusion

-        termed as SFPFusion  termed SFPFusion

-        state-of-the-arts  state-of-the-art

5)     Please observe the spacing/no spacing between characters throughout the article; for example: Transformer[42]  Transformer [42], …

Round 2

Reviewer 1 Report

Most of comments have been considered by authors in the revised version. The revised version is better than original submission in terms of paper organization and technical details. 

Reviewer 2 Report

I have no further comment.